# Increasing the Representation of Women in Diabetes Research

Kirsten Riches-Suman 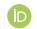

Cardiovascular Research Group, Faculty of Life Sciences, University of Bradford, Bradford BD7 1DP, UK;
k.riches@bradford.ac.uk; Tel.: +44-1274-232145

**Abstract:** Approximately half of all people with diabetes are women; however, the inclusion of women at all levels of research on diabetes is lacking. Clinical and pre-clinical trials do not have gender equity despite the differing progression of diabetes complications in women, and fewer women sit in academic or clinical leadership positions in diabetes than men. Whilst this scenario is not unique to diabetes, the purpose of this opinion article is to evaluate women's position in diabetology and focus on why the drive for gender equity at all levels is important. This article serves as a preface to the upcoming Diabetology Women's Special Issue Series, which aims to highlight and celebrate the achievements of women and people who identify as women in the hope of raising female voices in diabetes research and practice.

**Keywords:** type 1 diabetes; type 2 diabetes; gestational diabetes; gender differences

## 1. Introduction

The prevalence of diabetes mellitus is growing worldwide, with recent studies estimating that 536.6 million people (or 10.5% of the global population) currently have the disease [1]. There is variability regarding the prevalence of type 1 and type 2 diabetes (T1DM and T2DM, respectively) between men and women according to the global region in question; however, overall, the proportions of women and men being diagnosed with diabetes are comparable [1–11] (Figure 1). Gestational diabetes (GDM) presents exclusively in pregnant individuals. This commentary will briefly evaluate the current inequities in the representation of women in the study and leadership of diabetes research and practice, with the hope that this will identify areas for improvement for what is undoubtably a complex, multi-factorial and wide-ranging issue.

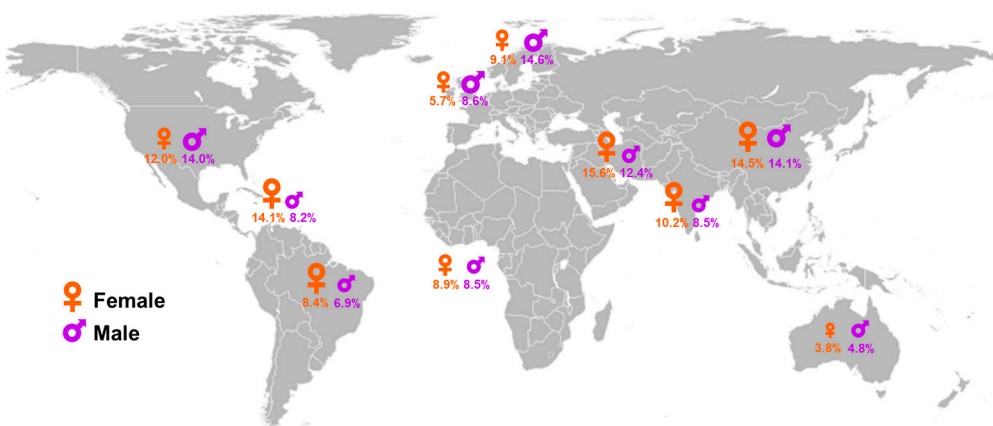

**Figure 1.** Prevalence of diabetes by gender. Whilst the prevalence of diabetes globally is comparable between men and women, there are regional differences. Data collated from literature presented from ten distinct global locations [2–11].

## 2. Differences in the Clinical Journey of Diabetes among Women

Whilst women and men are diagnosed with diabetes on an equal level overall, there are considerable variances in the presentation of the disease and its commonly associated complications. Non-genetic factors such as low socioeconomic status and lower levels of education are greater risk factors for developing diabetes in women compared to men [12,13]. It is currently unclear why this happens, but it may reflect cultural or social constructs which reduce women's likelihood of seeking medical attention and being diagnosed or the differential effects of lifestyle factors, such as smoking, on female bodies [12]. Adult women tend to be diagnosed with diabetes later in life [11], when the disease may have progressed further without interventions for glycemic control, than men. This is particularly important because of the importance of maintaining good glycemic control to manage the vascular complications of diabetes [14].

The leading cause of morbidity and mortality in patients with diabetes has traditionally been considered to be cardiovascular disease (CVD); however, a recent large-scale study suggested that in light of the significant improvements in cardiovascular therapies over the last two decades, the evidence now points to cancer as the leading cause of death of diabetes patients [15]. Nevertheless, cardiovascular morbidity and the number of patients requiring cardiovascular care due to diabetes are still high. Women are protected against CVD in their pre-menopausal years, with the onset of clinical atherosclerosis being delayed by approximately 10 years. However, women lose some, if not all, of this protection, particularly against coronary heart disease, when they acquire any type of diabetes [16].

Women with T1DM have a doubled risk of vascular events compared to men with T1DM and a 40% higher risk of all-cause mortality [17]. With T2DM, women have almost twice as much risk of developing coronary heart disease than men with T2DM and a 27% higher risk of stroke [18]. Whilst GDM is typically resolved upon delivery of the baby, women who have experienced GDM are much more likely to develop T2DM within the subsequent few years [19] and become susceptible to the associated cardiovascular morbidity and mortality [20].

The reasons for these differences in the severity and risk of cardiovascular diseases are complex and not clearly defined. For example, there are gender-specific differences in the risk factors for developing T2DM, including adipose distribution, hormones, stress responses and work patterns (thoroughly reviewed by the authors of [12]). These gendered differences in risk factors and clinical manifestations highlight the importance of having equal gender representation in both basic and clinical research. The upcoming Diabetology Special Issue on gender differences in diabetes will prove to be critical reading for anyone interested in learning more about these issues.

## 3. Representation of Female Biological Processes in Basic and Clinical Diabetes Research

Given these differences in the manifestation of diabetes complications, the ideal scenario would be for women to be represented in research studies (clinical or pre-clinical) equally to men. The involvement of women in clinical trials has improved over time, yet the question of how much equity in representation has been achieved remains contentious. Whilst some studies report that women now represent nearly 50% of clinical trial participants for all conditions globally [21,22], other studies reporting on female involvement cite much lower values that are more akin to 20% in different phases of clinical trials [23]. There is also variability across different clinical fields. In 2010, the representation of women in diabetes clinical trials was 40%, which is encouraging. However, given that coronary heart disease is a grave clinical concern for women with diabetes, women are still woefully under-represented in cardiology clinical trials [24].

This lack of representation may have significant impacts on the generation of new therapies that are effective in women. It is well known that male and female physiology and pharmacology differ. The inadequate representation of women in clinical trials of new diabetes and cardiovascular therapies can unintentionally lead to treatments that are effective in men but not necessarily as effective in women or have more adverse effects in women [25,26].

This Special Issue extends its scope to pre-clinical and basic research. Animal models of diabetes, whether they are obtained via chemical induction, genetic manipulation or diet-control, differ in terms of presentation and susceptibility between genders [27]. Indeed, murine and rodent models of diabetes are by far the most frequently used animal models; however, many female animals exhibit resistance to the development of diabetes and, as such, are not utilized in experimental groups which, in addition to the complexities of female hormonal cycles, has historically led to the exclusion of female animals from research. Large animal models such as porcine or non-human primates are far more representative of the presentation and complications of diabetes; however, the increased costs and timelines required to use these models with sufficient power precludes their frequent use [27] (Figure 2). Given our growing understanding of how gender impacts on the presentation and complications of diabetes, this historical exclusion of female animals is now changing, with growing calls for equal numbers of male and female animals of all types being used in all basic research pertaining to human disease [28].

**Diabetes patients**

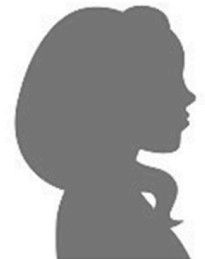

- **Average age of diagnosis in females is later than males**
- **There is an equal prevalence of diabetes across gender**

Women have an increased susceptibility to coronary heart disease, stroke and vascular disease

**Rodent and murine models of diabetes**

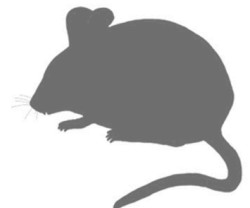

- **Many female rodent and murine models are resistant to diabetes development**
- **Studies carried out more frequently in male animals**

Female db/db mice more resistant to stroke

**Porcine models of diabetes**

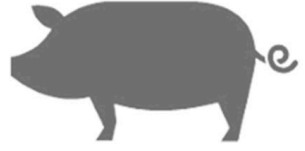

- **Many porcine models of diabetes present similarly between genders**
- **Good model for gestational diabetes**

Female Ossabaw pigs have greater susceptibility to coronary artery disease

**Figure 2.** Differences in the presentation of diabetes complications between male and female bodies. Women are equally as likely to be affected by diabetes as men and have a greater number of cardiovascular complications. In contrast, female small animal models of diabetes are relatively resistant to disease induction compared to male animals. Larger animal models are more representative of the female human presentation but are infrequently used. Please see the text for supporting references.

## 4. Women in Academic Leadership Roles

The lack of female representation of diabetes necessitates strong female leadership and, indeed, strong leadership from men who are fully aware of gender issues within diabetes research and care and who are committed to equitable change. This is important not only in clinical practice but also within academia. However, recent studies and commentaries examining the proportion of female representation in academic diabetes leadership have produced stark results. In terms of basic research, female voices are lacking. As with many other disciplines, diabetes research associations offer prestigious medals to outstanding scientists. Unfortunately, a recent commentary highlighted how only between 6 and 15% of prestigious research prize awardees in diabetes were women [29,30]. Furthermore, female leads of learned societies, such as the American Diabetes Association Presidents of Medicine and Science, account for only 11% of the presidents [30], and female voices occupy just over a quarter of editorial positions in leading diabetes journals, which drops to just 16% for senior editorial positions [29]. Of even greater concern is the fact that this

trend has not changed over time, demonstrating that ongoing initiatives to improve female representation in STEM have not yet had an impact on the field of diabetes research [30]. This finding is supported by a very basic analysis of people presenting as women in academic leadership positions on a professional networking site (Figure 3).

This is admittedly a very blunt and rudimentary tool for examining a complex and widespread issue. However, it is exactly the type of tool that the next generation of diabetes researchers will be using when they are choosing their career paths. If women, or people who identify as women, are not publicly visible in their chosen fields, this will perpetuate the sense of 'I do not belong there' which contributes in part towards the continuing attrition of women in STEM at higher professional levels. Given increasing societal awareness, it is refreshing to see positive initiatives for improving equity in diabetes research [31,32], which raise hope for a more balanced research voice in the future.

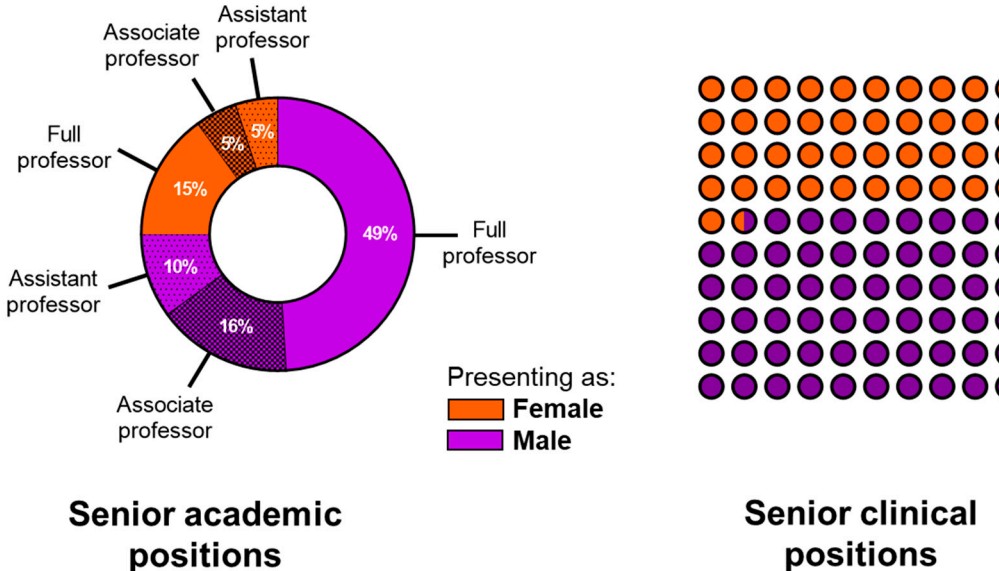

**Figure 3.** Leadership gender balance in diabetes research and clinical care. Left-hand panel: Presented gender balance of individuals found on LinkedIn using the search terms 'professor' and 'diabetes'. The resultant list was filtered to include only those individuals for whom diabetes was listed in their job title, location (e.g., a diabetes research institute) or skillset. Search performed on 6 February 2023. Right-hand panel: The average proportion of practicing diabetes and endocrinology consultants across the UK and USA. Data collected from the proportion of women listed in reference [33].

The picture in clinical practice is slightly more balanced. In the UK, women comprise 34% of diabetes and endocrinology consultants; this represents a near-doubling of the proportion compared to ten years prior. In the USA, the number is higher, at 49%, and women are over-represented at resident and fellow levels [33] (Figure 3). Whilst on the surface this appears to be a win for gender equity, it has been surmised that the proportional increase in female representation in endocrinology clinics is not due to an increase in female applications and appointments but rather due to a decrease and disinterest in the field by men, resulting in a proportional swing [34].

Increased female leadership necessitates equity in the amount of funding being awarded to people who identify as women, as leadership is intimately linked to career progression. Dunne et al. (2021) highlighted that men are twice as likely to receive diabetes research funding than women across a number of schemes, which could be attributed to the fact that less women are applying for the funding schemes in the first place [30]. The question of why women are less likely to apply for funding requires careful consideration and is beyond the scope of this article. Factors including unconscious (or indeed conscious) bias, the over-burdening of women with administrative or pastoral roles, reticence in promoting our own ideas and the lack of retention of women in STEM are all undoubt-

edly involved. However, funding bodies are becoming more aware of these issues and are starting to propose new policies and procedures to remove inequities in funding for female researchers [29].

### 5. Conclusions

The experience of diabetes among women is different from that of men, particularly in the presentation of cardiovascular disease. This is a significant problem, given that approximately half of all diabetes patients are female, amounting to ~268,500,000 people across the globe (diabetesatlas.org). Therefore, it is clear that basic and clinical research needs to reflect this trend so as to ensure the timely development of new therapeutics that improve health and well-being for all. However, equity in gender representation at every level of research needs to be improved. One way to achieve this is to raise the volume of the female voices in the academic and clinical management of diabetes. Initiatives such as the current Special Issue, dedicated to celebrating the achievements of women in diabetology, can contribute a small way towards this goal, but greater societal change is needed to achieve full equity.

Looking to the future, this is only the beginning of measures for ensuring equitable research and clinical practices across the board, and not only for diabetes. Gender is an important part of how bodies respond to disease, and our improving recognition and acceptance of alternative genders including (but not limited to) individuals who are transgender or intersex will necessitate and, indeed, demand research regarding how hormonal and physiological changes in a persons' body and the timing of these changes affect the development, presentation and complications of diabetes. Hopefully, raising female voices in diabetology will pave the way for other minoritized groups to achieve equity in diabetes research and leadership.

**Funding:** This research received no external funding.

**Institutional Review Board Statement:** Not applicable.

**Informed Consent Statement:** Not applicable.

**Data Availability Statement:** Data are available upon request.

**Conflicts of Interest:** The author declares no conflict of interest.

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
