# Peer review of "Increasing the Representation of Women in Diabetes Research"

_diabetology, doi:10.3390/diabetology4020014_

Round 1

Reviewer 1 Report

The commentary by Riches-Suman is nicely written preface for upcoming Diabetology Women’s Special Issue series. However, I have a couple of suggestions/ questions: 

1.     It will be beneficial to provide more statistics in Introduction section such as number of diabetes patients globally and further divide in male vs females. 

2.     Line 31, “women lose some, if not all”. If available, can you include percentage change or fold change in the protection. 

3.     Heading of Section 3 can be basic and clinical diabetes research. 

4.     In section 3, you presented data about gender disparity in clinical trials. Does similar data exist for animal studies? If yes, it be beneficial to include disparity in basic science research. 

5.     Lines 78-80, again, it will be beneficial to provide statistical data to shown how females are lacking in editorial board and awards. 

6.     Is there data available about involvement of females in funding agencies and policy making? Does funding agencies emphasis or put forward any criterion which minimize gender disparities in research. 

7.     In figure 1, elaborate on senior clinical positions.

8.     It will be beneficial to briefly talk about effect of socioeconomic status and/or education. The idea is that if women are aware of resources available. 

Author Response

I thank the reviewer for their thoughtful and constructive critique and have responded to all points below.

The commentary by Riches-Suman is nicely written preface for upcoming Diabetology Women’s Special Issue series. However, I have a couple of suggestions/ questions:

  1. It will be beneficial to provide more statistics in Introduction section such as number of diabetes patients globally and further divide in male vs females.

Thank you for this suggestion. I have added the number of cases of diabetes globally in the text (line 20) and have created an infographic that summarises the prevalence of diabetes stratified by gender in ten different locales around the world. This demonstrates visually that although the global picture is of an equal impact of diabetes on men and women, there can be large differences according to region of the world in which you live.

  1. Line 31, “women lose some, if not all”. If available, can you include percentage change or fold change in the protection.

The fold changes for this statement are not available. However I have added extra detail in that women typically develop clinically-impactful atherosclerosis ten years later than men. The subsequent paragraph (current line 56-62) detail more specific risks for the distinct types of diabetes and different cardiovascular outcomes.

  1. Heading of Section 3 can be basic and clinical diabetes research.

The change has been made as requested.

  1. In section 3, you presented data about gender disparity in clinical trials. Does similar data exist for animal studies? If yes, it be beneficial to include disparity in basic science research.

In the final paragraph of this section (current lines 90-103), I refer the reader to a review article examining the historical lack of female murine research, the reasons behind this, and the growing call to action to improve the gender disparity in animal research into diabetes. I could not find any sources that quantified the number of studies which were done exclusively on male animals and thus rely on the perspectives presented in the original reference (Daniels-Gatward et al, 2021). From my own experience, absolute quantifications are quite difficult to attain as primary research articles do not always include the gender split of their animals in the methods section. I have now expanded this section to include an article that looks at the differing development and presentation of diabetes in other large animal models to emphasise that gender-stratified studies are needed and have included an infographic highlighting the disparities.

  1. Lines 78-80, again, it will be beneficial to provide statistical data to shown how females are lacking in editorial board and awards.

I have expanded the text in this section to be more explicit about the numbers provided in the cited references.

  1. Is there data available about involvement of females in funding agencies and policy making? Does funding agencies emphasis or put forward any criterion which minimize gender disparities in research.

I have added an extra paragraph highlighting studies that have looked at inequities in research funding. Some funding agencies such as the European Research Council are starting to implement equity policies however these are still in their infancy.

  1. In figure 1, elaborate on senior clinical positions.

Thank you for this question. Senior clinical positions refers to consultants in the UK or practicing endocrinologists in the USA. I have added the word ‘practicing’ to the figure legend to help clarify this.

  1. It will be beneficial to briefly talk about effect of socioeconomic status and/or education. The idea is that if women are aware of resources available.

Thank you for this suggestion. I have added new text addressing this issue on lines 38-45.

Reviewer 2 Report

I think that this opinion document is summarized as follows; first, although differential clinical course and prognosis in diabetes between men and women, cause of diabetic women's disease may not be fully evaluated. Second, diabetic research must be more open to women's researchers. 

 I am agreed with the above points. Inline 28-29, author wrote that the leading cause of morbidity and mortality in patients with diabetes is cardiovascular disease (CVD). 

 However, recently, Jonathan Pearson-Stuttard et al reports that vascular death rates has been declined in diabetes and the leading contributor to diabetes-related death was changed from CVD to cancer( Lancet Diabetes Endocrinol 2021; 9: 165–73).  Hence, although CVD is still important problem, such as cancer and dementia are also recently important cause of death in diabetic patients. Therefore, author must check and mention recent cause of death in diabetes before discuss CVD in diabetes. 

Please clarify.

Author Response

I thank the reviewer for their time and comments, and have addressed their concerns below.

I think that this opinion document is summarized as follows; first, although differential clinical course and prognosis in diabetes between men and women, cause of diabetic women's disease may not be fully evaluated. Second, diabetic research must be more open to women's researchers. I am agreed with the above points. Inline 28-29, author wrote that the leading cause of morbidity and mortality in patients with diabetes is cardiovascular disease (CVD). However, recently, Jonathan Pearson-Stuttard et al reports that vascular death rates has been declined in diabetes and the leading contributor to diabetes-related death was changed from CVD to cancer( Lancet Diabetes Endocrinol 2021; 9: 165–73).  Hence, although CVD is still important problem, such as cancer and dementia are also recently important cause of death in diabetic patients. Therefore, author must check and mention recent cause of death in diabetes before discuss CVD in diabetes. Please clarify.

Thank you for raising my awareness of this important paper. I have now included text that refers to cancer now being the leading cause of death (and this citation), but that the number of patients seeking cardiovascular treatments due to diabetes is still a significant clinical problem.

Reviewer 3 Report

no comment needed, the paper is concise and clear

Author Response

Thank you for your positive evaluation